# *LjAMT2;2* Promotes Ammonium Nitrogen Transport during Arbuscular Mycorrhizal Fungi Symbiosis in *Lotus japonicus*

**DOI:** 10.3390/ijms23179522

**Published:** 2022-08-23

**Authors:** Yanping Wang, Wenqing Zhou, Jiandong Wu, Kailing Xie, Xiaoyu Li

**Affiliations:** National Engineering Laboratory of Crop Stress Resistance Breeding, Anhui Agricultural University, Hefei 230036, China

**Keywords:** arbuscular mycorrhizal fungi, ammonium transporter, AMT, ammonium nitrogen, symbiosis, *Lotus japonicus*

## Abstract

Arbuscular mycorrhizal fungi (AMF) are important symbiotic microorganisms in soil that engage in symbiotic relationships with legumes, resulting in mycorrhizal symbiosis. Establishment of strong symbiotic relationships between AMF and legumes promotes the absorption of nitrogen by plants. Ammonium nitrogen can be directly utilised by plants following ammonium transport, but there are few reports on ammonium transporters (AMTs) promoting ammonium nitrogen transport during AM symbiosis. *Lotus japonicus* is a typical legume model plant that hosts AMF. In this study, we analysed the characteristics of the *Lotus japonicus* ammonium transporter *LjAMT2;2*, and found that it is a typical ammonium transporter with mycorrhizal-induced and ammonium nitrogen transport-related cis-acting elements in its promoter region. *LjAMT2;2* facilitated ammonium transfer in yeast mutant supplement experiments. In the presence of different nitrogen concentrations, the *LjAMT2;2* gene was significantly upregulated following inoculation with AMF, and induced by low nitrogen. Overexpression of *LjAMT2;2* increased the absorption of ammonium nitrogen, resulting in doubling of nitrogen content in leaves and roots, thus alleviating nitrogen stress and promoting plant growth.

## 1. Introduction

Arbuscular mycorrhizal fungi (AMF) are monophyletic saccular fungi that can form a mutually beneficial mycorrhizal symbiosis with 70–90% of vascular plants on land [1]. The establishment of AM symbiosis plays an important role in the balance of carbon and nitrogen in ecosystems [1,2,3,4]. Mycorrhizal plants uptake nutrient elements in two ways: directly through plant roots, or indirectly through a symbiotic pathway. Tracer studies have shown that nitrogen transfer also occurs between fungal symbiotes and hosts, and up to 42% of nitrogen can be obtained through mycorrhizal symbiosis [5].

In most soils, the concentration of ammonium nitrogen (1–25 μM) is typically lower than that of nitrate nitrogen (100 μM to 70 mM) [6]. However, because NH4^+^ can be directly assimilated via the GS/GOGAT pathway, ammonium nitrogen plays a key role in plant symbiosis. The amount of nitrogen transported by AMF to maize plants depends on the form of nitrogen; the rate of ammonium ion transfer by maize mycorrhiza is 10 times higher than that of nitrate ions [7,8]. Symbionts promote ammonium nitrogen uptake in plants mainly through regulation of related genes. Ammonium nitrogen is transported into plants through ammonium transporters (AMTs) on the plasma membrane of root cells [9]. These carrier proteins mediate NH4^+^ transport across plant membranes [10].

In angiosperms, ammonium nitrogen transporters are divided into two families: AMT1 and AMT2 [11,12]. High-affinity AMT1 subfamily genes are the main AMTs that respond to nitrogen starvation stress. For example, expression of *AtAMT1;1* and *AtAMT1;3* in *Arabidopsis thaliana* roots was significantly increased after low-nitrogen treatment [13]. *PyAMT1* in *Pyropia yezoensis* is strongly upregulated by low nitrogen, and it plays an important role in nitrogen assimilation [14]. *ZmAMT1;1a* and *ZmAMT1;3* in maize are continuously induced by ammonium nitrogen, and they can supplement the functions of ammonium nitrogen transport-deficient mutants and promote the absorption of ammonium nitrogen [15].

In contrast, expression of ATM2 genes often seems to be induced by AMF. Although it is not clear how plants absorb ammonium nitrogen released by AMF, numerous plant ammonium transporters induced by AMF have been identified in various plants, and they may be involved in the transfer of ammonium nitrogen from AMF. For example, *MtAMT2;2* in *Medicago truncatula* was specifically upregulated during mycorrhizal symbiosis [16]. *GmAMT4.1* in soybean and *SbAMT3.1* in sorghum were confirmed to be located on the membrane around arbuscular branches, and *SbAMT4* could complement ammonium-deficient yeast mutants [16,17,18]. *LeAMT4* and *LeAMT5* in tomato were only expressed in mycorrhiza [19]. Heatmap analysis of transcriptome data from *Lotus japonicus* under symbiosis formation for 4 and 28 days in non-symbiotic conditions showed that expression of AMT1 subfamily genes was hardly affected by AMF [20,21,22]. By contrast, AMF had a significant effect on expression of AMT2 family members, consistent with previous reports on AMTs induced by AMF in other species [16,17,18,20], among which the *LjAMT2;2* gene was most significantly affected by AMF. Indeed, it has been reported that in the transcriptomic analysis of *Lotus japonicus* roots after AMF colonization, *LjAMT2;2* was shown to be the most upregulated gene, as the transcript was preferentially located in arbuscular cells and had the function of transporting NH_3_, which is the key to AMT in mycorrhiza [20].

In the present work, the characteristic analyses of *LjAMT2;2* indicate that *LjAMT2;2* is a typical ammonium transporter, with both mycorrhizal-induced and ammonium nitrogen transport-related cis-acting elements in its promoter region. *LjAMT2;2* facilitated ammonium transfer in yeast mutant supplement experiments. Overexpression of *LjAMT2;2* increased the absorption of ammonium nitrogen, resulting in the doubling of nitrogen content in leaves and roots, thus alleviating nitrogen stress and promoting plant growth.

## 2. Results

### 2.1. Bioinformatics Analysis of the LjAMT2;2 Gene

AMT genes in *Lotus japonicus* were compared with their counterparts in maize, rice, *A. thaliana*, soybean and sorghum using MEGA7 software to construct a phylogenetic tree (Figure 1A). Among these genes, *ZmAMT3;1*, *SbAMT3;1*, *SbAMT4* and *GmAMT4;1* are all reported to be upregulated by AMF. In the tree, AMTs are mainly divided into two subfamilies in plants, and genes of the AMT2 subfamily are induced by AMF. *LjAMT2;2* was closely related to *SbAMT4* and *GmAMT4;1*, suggesting that the expression of *LjAMT2;2* is also specifically induced by AMF.

To further determine the similarity between *LjAMT2;2* and other reported AMTs induced by AMF, amino acid sequences of LjAMT2;2, SbAMT4, GmAMT4;1, ZmAMT3;1 and SbAMT3;1 were compared using ClustalX and Boxshade (Figure 1B). The results showed that LjAMT2;2 possesses multiple ammonium transporter domains that are highly conserved. Comparison of sequences with DMAMAN software showed that the amino acid sequence identity was as high as 73.88%.

The three-dimensional structure of the protein was visualised using Pymol software (Figure 1C). The LjAMT2;2 protein has the typical homotrimer structure of members of the ammonium transporter family, and each monomer has a hydrophobic pore in the centre. The amino acid sequence of LjAMT2;2 was analysed using TMHMM to predict possible transmembrane domains, and 11 transmembrane helices were predicted (Figure 1D).

Together, the results of bioinformatics analyses suggested that *LjAMT2;2* was specifically induced by AMF and assisted in transporting ammonium.

### 2.2. LjAMT2;2 Subcellular Localisation

*A. tumefaciens* cells separately transformed with an empty vector and *LjAMT2;2*-GFP were injected into tobacco leaves and cultured in the absence of light for two days. The fluorescence of leaves was then observed under a confocal microscope. In the empty vector control group, fluorescence was observed in the plasma membrane and nucleus, while green fluorescence was only observed on the plasma membrane in the group expressing the LjAMT2;2 protein, suggesting that it is located on the plasma membrane (Figure 2).

### 2.3. Functional Analysis of LjAMT2;2 in Ammonium-Deficient Yeast

In order to verify the ammonium transport function of LjAMT2;2, the ammonium transporter yeast strain 31019b (mep1 Δ, mep2 Δ:: LEU2,mep3 Δ:: KanMX2 ura3) was used. These cells could not grow normally when ammonium was used as the sole nitrogen source and when the concentration of ammonium ion was <5 mM. The yeast supplementation experiment showed that in the YNB medium containing 2 mM arginine (0.17% YNB + 2% D-galactose + 2 mM arginine + 2% agar), cells transformed with pYES2 or *LjAMT2;2*-pYES2 vectors grew normally. However, in the YNB medium with an ammonium ion concentration of 0.02 mM, 0.2 mM or 2 mM, cells transformed with pYES2 could not grow normally, while cells transformed with *LjAMT2;2*-pYES2 could grow normally, and growth was stronger with increasing ammonium ion concentration. This indicates that the *LjAMT2;2* gene can complement the phenotype of ammonium transporter-deficient yeast, suggesting a function in ammonium translocation (Figure 3A). In 2009, Guether et al. reported that *LjAMT2;4* was significantly upregulated by AM fungi at 28 dpi, so the yeast growth curve was also measured. Yeast growth curves showed that cells transformed with *AtAMT1;2*-pYES2 as a positive control began to grow exponentially after 20 h of culture. Compared with the control group, cells transformed with *LjAMT2;2*-pYES2 grew more rapidly after 24 h, then slowly entered a steady phase after 70 h. These results further indicate that *LjAMT2;2* could complement the phenotype of ammonium-deficient yeast, suggesting its important function in ammonium transport. Meanwhile, our results showed another AMT2 gene in *Lotus japonicus*, *LjAMT2;4*, had limited function in transporting ammonium nitrogen (Figure 3B).

### 2.4. Effects of AMF Colonisation on Lotus japonicus Growth under Different Ammonium Concentrations

We explored the effects of AMF colonisation on the growth of *Lotus japonicus* under different ammonium ion concentrations. The results showed that the symbiosis rate between roots and AMF reached more than 50%, and the mycorrhizal colonisation rate increased with increasing ammonium ion concentration. When the concentration was 1 mM, the mycorrhizal colonization rate was highest (Figure 4A), and most roots achieved symbiosis. Subsequently, we assessed aboveground and underground parts of roots under different treatments (Figure 4B,C). The results showed that the aboveground and underground biomasses of roots with AMF were significantly higher than without AMF. Thus, under nitrogen starvation conditions, the dominant role of AMF was more significant.

### 2.5. Mycorrhizal-Inducible Expression of the LjAMT2;2 Gene

We then sought to verify that the *LjAMT2;2* gene is specifically induced by AMF. A previous study investigated the AMF transcriptome following symbiosis with *Lotus japonicus* by comparing transcriptome data for *Lotus japonicus* at 4 and 28 days of symbiosis with data acquired under non-symbiotic conditions [20]. The heatmap showed that expression of the *LjAMT2;2* gene was altered significantly under the influence of AMF; expression of *LjAMT2;2* was upregulated by AMF at 28 days of symbiosis (Figure 5A).

Next, we verified the AMF induction experiment further (Figure 5B–D). Under AM symbiosis, roots of plants transformed with p*LjAMT2;2*-GUS were stained blue by X-Gluc dye, indicating GUS expression, whereas roots without AMF were not stained blue (Figure 5B). Microscopy observation of symbiotic roots showed that GUS expression was mainly in the vascular pillar and endothelial cells near the vascular column, with lower expression levels in the outer layer cells (Figure 5B).

In order further confirm that the *LjAMT2;2* was induced by AMF, the promoter sequence of the gene was further analysed. The promoter regions of AMF-induced genes generally have cis-acting elements related to mycorrhizal induction, such as W-BOX (TTGACC) and OSEROOTNODULE (AAAGAT) [23]. The sequence 2000 bp upstream of the *LjAMT2;2* gene was analysed using the RSAT promoter online tool. The results revealed numerous cis-acting elements in the promoter region of the *LjAMT2;2* gene, and four cis-acting E-BOX elements related to ammonium nitrogen transport in the promoter region of *LjAMT2;2* (Figure 5C).

Then, the effect of AMF colonisation on expression of the *LjAMT2;2* gene in roots under different ammonium ion concentrations was studied (Figure 5D). The results showed that with all ammonium ion concentrations tested, *LjAMT2;2* was barely expressed in roots under non-symbiotic conditions, but expression was significantly upregulated in symbiotic mycorrhiza. Furthermore, analysis revealed that at an ammonium ion concentration of 0.5 mM, expression of *LjAMT2;2* in root without AMF was highest; this was the most suitable ammonium ion concentration, and the general trend showed a normal distribution. After inoculation with AMF, expression of *LjAMT2;2* decreased with increasing ammonium ion concentration. We speculated that an increase in ammonium ion concentration might prevent the occurrence of ammonium toxicity and decrease gene expression. At the lowest ammonium ion concentration (0.05 mM), expression of the *LjAMT2;2* gene with AM symbiosis was more than 200 times higher than that without AM symbiosis, indicating that the gene was significantly upregulated by AMF in a low-nitrogen environment.

### 2.6. Phenotypic Identification and Physiological Index Analysis of LjAMT2;2-OE Plants

Construction of overexpression vectors and identification of overexpression plants are shown in Appendix A. Phenotypic identification was performed on *LjAMT2;2*-OE and wild-type plants cultured for eight weeks (Figure 6A,B). The results showed that when the concentration of ammonium nitrogen was low (0.05 mM), the aboveground biomass, underground biomass, root length and lateral root number of *LjAMT2;2*-OE plants was increased by 124.7%, 39.75%, 9.8% and 78.05%, respectively, compared with wild-type plants. When the concentration of ammonium nitrogen was only slightly deficient (0.5 mM), the aboveground biomass was increased by 137.5%, the underground biomass was increased by 119.33%, the root length was increased by 6.67%, and the number of lateral roots was increased by 55.36%. Therefore, *LjAMT2;2*-OE plants showed increased growth and development, and absorption of ammonium nitrogen was enhanced to a certain extent. Under nitrogen-deficient conditions, root length was increased to absorb more ammonium nitrogen to meet growth needs. When nitrogen deficiency was relaxed, growth of lateral roots was stimulated.

The total nitrogen content in aboveground parts of wild-type and *LjAMT2;2*-OE plants under different treatments was determined (Figure 7A). At both 0.05 mM and 0.5 mM ammonium ion concentrations, the nitrogen content of overexpression plants was significantly higher than that of wild-type plants, especially under extreme nitrogen starvation conditions, indicating that absorption of ammonium nitrogen by roots was enhanced in *LjAMT2;2*-OE plants, especially in a low-nitrogen environment.

The MDA content indirectly reflects nitrogen starvation stress. Thus, we determined MDA levels in aboveground parts and found that levels in *LjAMT2;2*-OE plants were significantly lower than in wild-type plants (Figure 7B). We therefore speculated that LjAMT2;2 alleviates the symptoms of nitrogen deficiency in plants to some extent, thereby reducing stress in plant cells, hence the MDA content decreases.

## 3. Discussion

AMF are important symbiotic microorganisms in soil that form arbuscular mycorrhiza with host plants. After establishing a stable symbiotic relationship with plants, AMF exchange nutrients with plants through arbuscular branches and hyphae through a mutually beneficial relationship [24], and in return plants provide carbohydrates to AMF in the form of lipids and sugars [2,25,26,27,28]. Furthermore, AMF transport essential nutrients such as nitrogen and water to plants.

The main purpose of this study was to investigate the effect of AMF on the function of members of the AMT family. The LjAMT2;2 protein contains 11 transmembrane domains, and conserved domains of ammonium transporters in the fifth to sixth transmembrane domains, consistent with the characteristics of typical ammonium transporters [29]. Organic nitrogen absorbed by plants can be divided into nitrate nitrogen and ammonium nitrogen, of which ammonium nitrogen does not need to be transformed, and is directly absorbed and utilised by plants. Ammonium nitrogen is mainly transported to plant cells through ammonium transporters (AMTs) on the cell membrane. A certain amount of ammonium in soil (usually < 100 μM) is beneficial to plant growth, while excess ammonium is toxic [30]. Therefore, it is necessary to strictly regulate ammonium absorption in roots, which is mainly mediated by regulating the expression and activity of AMTs. It has been reported that AMF help plants increase ammonium uptake by upregulating the expression of ammonium transporters, as demonstrated for *LeAMT4* and *LeAMT5* in tomato, *SbAMT3;1* in sorghum and *GmAMT4.1* in soybean are essential players in the nitrogen-regulation mechanism [17,18,19].

To explore whether the protein functions in ammonium transport, the ammonium-deficient yeast strain 31019b was supplemented with each construct, and the results showed that expressing *LjAMT2;2* could restore the growth of defective yeast cells grown on NH4^+^ single nitrogen source medium, indicating that LjAMT2;2 functions in ammonium transport (Figure 3).

To confirm that the *LjAMT2;2* gene was indeed induced by AMF, wild-type plants were treated with different concentrations of single ammonium nitrogen nutrient solution. The experimental group was inoculated with AMF and the control group was not inoculated with AMF. The results showed at all ammonium ion concentrations tested, expression of *LjAMT2;2* was significantly upregulated by AMF (Figure 5D). GUS staining showed that the protein was located on the arbuscular branches near the vascular column, consistent with GUS activity in endothelial cells in previous studies 33, indicating that the promoter of the gene was induced by AMF (Figure 5B). This is consistent with the work of Yoshihiro Kobae et al. in 2010 [18]. Cells harbouring p*GmAMT4;1*-GUS displayed consistent GUS activity in endodermis cells [18], indicating that the promoter of this gene was induced by AMF (Figure 5B).

We also found that with increasing ammonium ion concentration, the effect of AMF on *LjAMT2;2* expression gradually decreased; expression was upregulated most notably when nitrogen starvation was severe, indicating that the gene responds to low-nitrogen stress (Figure 4C). To investigate further whether the gene was induced by AMF, we analysed its promoter and found that the promoter region contains several cis-acting elements and nitrogen transport cis-acting elements that respond to AMF. The results of promoter element analysis confirmed that the gene might be induced by AMF (Figure 5C). *LjAMT2;2*-OE and wild-type plants were treated with 0.05 mM and 0.5 mM NH4^+^ single nitrogen source medium once a week for 8 weeks (wpi). The results showed that under severe nitrogen starvation, aboveground and underground biomasses of overexpressing lines were significantly increased, and lateral roots were also increased (Figure 6), but root length was slightly decreased. Consistently, under nitrogen deficiency, plants are known to obtain more nitrogen nutrition by increasing their root length and decreasing lateral roots [31]. Under mild nitrogen deficiency (0.5 mM ammonium), alleviation of nitrogen deficiency stress in *LjAMT2;2*-OE plants was not as obvious as that in plants under severe nitrogen deficiency, but it was also increased compared with wild-type plants. Overexpression of the *LjAMT2;2* gene alleviated the growth inhibition of aboveground parts under severe nitrogen deficiency, and this effect was relatively weak in an environment with sufficient ammonium nitrogen, presumably because plants were not starved of ammonium nitrogen.

To explore whether the *LjAMT2;2* gene promoted the absorption of ammonium nitrogen and alleviated stress in plants, we measured the total nitrogen content in aboveground parts of plants. The results showed that in *LjAMT2;2*-OE plants the nitrogen content in aboveground parts was significantly increased (Figure 7A), and the MDA content was decreased significantly compared with wild-type plants (Figure 7B). These results indicate that *LjAMT2;2* functions as an ammonium transporter in *Lotus japonicus*, and it plays a more important role under severe nitrogen stress conditions.

## 4. Materials and Methods

### 4.1. Bioinformatics Analysis

A phylogenetic tree was constructed using the neighbour-joining method in MEGA7 with self-expanding value = 1000 and Poisson model selected. A heatmap was drawn using the heat map function of TBtools [32]. The TMHMM website (http://cbs.dtu.dk/services/TMHMM/ (accessed on 20 Septemper 2019)) was used to predict transmembrane domains. Cis-acting elements were analysed in promoter regions within ~2000 bp upstream of genes using the RSAT website (http://floresta.eead.csic.es/rsat/ (accessed on 20 Septemper 2019)). ClustalX (https://www.genome.jp/tools-bin/clustalw (accessed on 25 Septemper 2019)) and Boxshade (https://embnet.vital-it.ch/software/BOX_form.html (accessed on 25 Septemper 2019)) were used for protein sequence alignment analysis.

### 4.2. Vector Construction

The promoter of *LjAMT2;2* (p*LjAMT2;2*) was used to drive GUS expression of pCAMBIA1301. *Hind*III and *Nco*I restriction enzyme sites were incorporated into forward primer 5′-GCAGGCATGCAAGCTTTCTCCAATCAAAAGGCACACC-3′ and reverse primer 5′-GCAGGCATGCAAGCTTTCTCCAATCAAAAGGCACACC-3′, and the PCR amplification product was ligated to generate the p*LjAMT2;2*-GUS plasmid. The subcellular localisation vector pCAMBIA1305-GFP and *Spe*I and *Bam*HI restriction sites were used with forward primer 5′-GACTAGTATGTCTACTGTTGTTCCACTTCC-3′ and reverse primer 5′-CGGGATCCTTAGTTTGAGGTCATCTCGAGC-3′ to generate the *LjAMT2;2*-GFP plasmid. The overexpression vector pCAMBIA1301-GFP and E*co*RI and *Hind*III restriction sites were used with forward primer 5′-GGGGTACCATGTCTACTGTTGTTCCACTTCC-3′ and reverse primer 5′-GCTCTAGATTAGTTTGAGGTCATCTCGAGC-3′ to generate the *LjAMT2;2*-OE plasmid. The yeast expression vector pYES2 and *Bam*HI and *Xba*I were restriction sites were used with forward primer 5′-CGGGATCCA TGTCTACTGTTGTTCCACTTCC3′ and reverse primer 5′-GCTCTAGATTAGTTTGAGGTCATCTCGAGC-3′.

### 4.3. Subcellular Localisation

Recombinant plasmid *LjAMT2;2*-GFP and the control vector were separately introduced into *Agrobacterium tumefaciens* strain GV3101, which was then injected into *Nicotiana benthamiana* leaves (at 5–6 weeks old). After injection, leaves were cultured in darkness for 48 h, and green fluorescent protein (GFP) fluorescence was observed using a laser confocal microscope (Leica LSM800, Weztlar, Germany) at 488 nm excitation.

### 4.4. Experimental Materials and Planting Treatments

The study material was *Lotus japonicus* wild-type MG20 and the AMF species was *G**lo**mus intraradices* (Gi, provided by Sun Yat-Sen University, Guangzhou, China). The sand/soil culture medium was mixed with vermiculite: perlite at a 4:1 ratio and sterilised by 40 min high-pressure steam at 121 °C. In order to verify whether *LjAMT2;2* was affected by AMF and to explore the response to ammonium concentration in symbiotic and non-symbiotic environments, four ammonium concentrations (0.05 mM, 0.5 mM, 1 mM and 4 mM) were tested to analyse symbiotic and non-symbiotic roots cultured for 28 days.

### 4.5. Transformation of Hairy Roots and Expression Induction by AMF

P*LjAMT2;2*-GUS was transferred into *A. tumefaciens* LBA9402 and cells were grown in YEB solid medium. The verified positive colonies were transferred to YMB solid medium and cultured at 28 °C for 2 days. During the culture of Agrobacterium tumefaciens, the seeds of *Lotus japonicus* were sterilized (seeds were sterilized with 12% NaClO for 10 min; washed with 75% ethanol for 3 times, 1–2 min each time; washed with sterile water 3–5 times, 4–5 min each time) and germinated. After 2 days of germination of *Lotus japonicus*, the hundred-pulse root was removed from the root tip to the root hair with an aseptic scalpel, and calli were infected with *A. tumefaciens* LBA9402 carrying p*LjAMT2;2*-GUS. Infected roots were transferred to B&D medium (Appendix A) and cultured in the dark at 28 °C for 24 h, then grown for ~3 weeks in a light incubator at 23 °C with an 8 h photoperiod before transplanting. After symbiotic culture with AMF for 6 weeks, expression of the GUS gene in roots was measured with staining at 37 °C for 24 h to identify whether the gene was induced by AMF.

### 4.6. Supplementation of Ammonium-Deficient Yeast

The ammonium transporter-deficient yeast strain 31019b (mep1Δ, mep2Δ:: LEU2, mep3Δ:: KanMX2 ura3) cannot grow normally when ammonium is used as the only nitrogen source and the ammonium ion concentration is <5 mM [33]. The pYES2-*LjAMT2;2* plasmid was transformed into ammonium-deficient yeast cells and cultured in YPD liquid medium until the optical density at 600 nm (OD600) was ~1.2 mL, and cells were collected and diluted 10:1, 10:2, 10:3 and 10:4. Samples were collected from YNB medium containing different ammonium ion concentrations (0.02 mM, 0.2 mM, 2 mM) or 2 mM arginine as a negative control. *AtAMT1;2* in *A. thaliana* [34], known to compensate the phenotype of ammonium transporter-deficient yeast, served as a positive control. Cells were cultured at 30 °C for 96 h.

To dynamically track the growth rate of yeast, cells were applied to YNB liquid medium containing 2 mM ammonium, the OD600 was adjusted to 0.2, culturing was performed with shaking. The OD600 value was measured at intervals, and the total culture time was 76 h. The growth rate of yeast was dynamically monitored.

### 4.7. RNA Isolation and Quantitative Real-Time RT-PCR (qPCR)

Total RNA was isolated using the phenol-chloroform extraction method. First-strand cDNAs were synthesised from DNaseI-treated total RNA using reverse transcriptase (TransGen, Beijing, China) and oligo-dT primers. To normalise gene expression, *ubiquitin* gene and *EFα1* genes were used as internal controls [35]. qPCR was conducted on an Applied Biosystems 7300 system using ChamQ Universal SYBR qPCR Master Mix (Vazyme, Nanjing, China) according to the manufacturer’s protocol. Expression levels in plants without inoculation of arbuscular fungi were used for normalisation, and relative gene expression levels were determined using the 10–(ΔCt/3) method [36]. All qPCR assays included three biological replicates.

### 4.8. Determination of Nitrogen and Malondialdehyde (MDA) Content

The nitrogen content in soil was determined by the sulphuric acid-accelerator digestion method [37]. Total nitrogen in soil was converted to ammonium nitrogen by redox reaction under the action of concentrated sulphuric acid and catalyst. The digested solution was alkalised, distilled ammonia was absorbed by boric acid, titrated with standard sulphuric acid solution, and the total nitrogen content in soil was calculated according to the amount of standard sulphuric acid solution. Total nitrogen in plants was digested with sulphuric acid and hydrogen peroxide to convert organic nitrogen into ammonium salts. Ammonium salts were alkalised to form ammonia, which was absorbed into boric acid solution by distillation. Using methyl red-bromocresol green as indicator, titration with standard acid was used to determine the total nitrogen content in plants.

The MDA content was determined using a Malondialdehyde Kit (BC0025, Solebao, Beijing, China). Overexpressing and wild-type *Lotus japonicus* plants were grown for 8 weeks under 0.05 mM or 0.5 mM ammonium, and biomass, root length, number of lateral roots, and nitrogen content and MDA levels in aboveground parts were measured.

## 5. Conclusions

LjAMT2;2 is a typical ammonium transporter with 11 transmembrane helices and highly conserved ammonium transporter domains. Its promoter contains numerous cis-acting elements related to mycorrhizal induction. In symbiotic mycorrhiza, *LjAMT2;2* was strongly upregulated by AMF, and induced by low nitrogen. *LjAMT2;2* supplemented the ammonium-deficient yeast mutant 31019b, indicating that the gene encodes a protein involved in ammonium translocation. Under nitrogen starvation conditions, *LjAMT2;2*-OE plants displayed significantly increased aboveground biomass, lateral root number, total root length and total nitrogen content, and lower MDA content, compared with wild-type plants. This indicates that overexpression of the *LjAMT2;2* gene promoted the growth and development of *Lotus japonicus* by stimulating ammonium absorption to alleviate nitrogen stress.

## Figures and Tables

**Figure 1 ijms-23-09522-f001:**
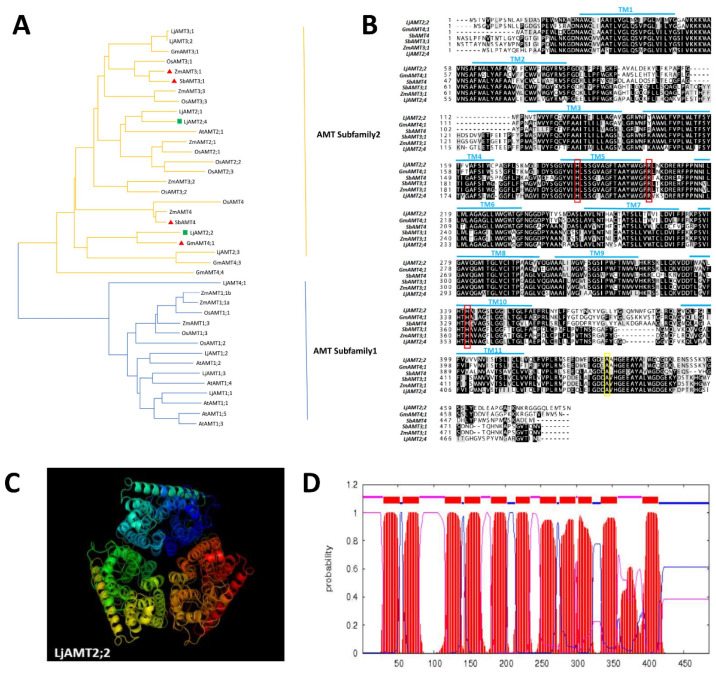
Bioinformatics analysis of the *LjAMT2;2* gene. (**A**) Evolutionary analysis of the LjAMT family and AMT families in other species. (**B**) Amino acid sequence alignment of LjAMT2;2 with other reported AMT genes. Red lines indicate conserved ammonium transporter domains, and yellow lines indicate highly conserved domains. (**C**) Predicted three-dimensional structure of the LjAMT2;2 protein. (**D**) Distribution of transmembrane domains in LjAMT2;2. Purple represents the inner membrane, red represents the transmembrane, and blue represents the outer membrane.

**Figure 2 ijms-23-09522-f002:**
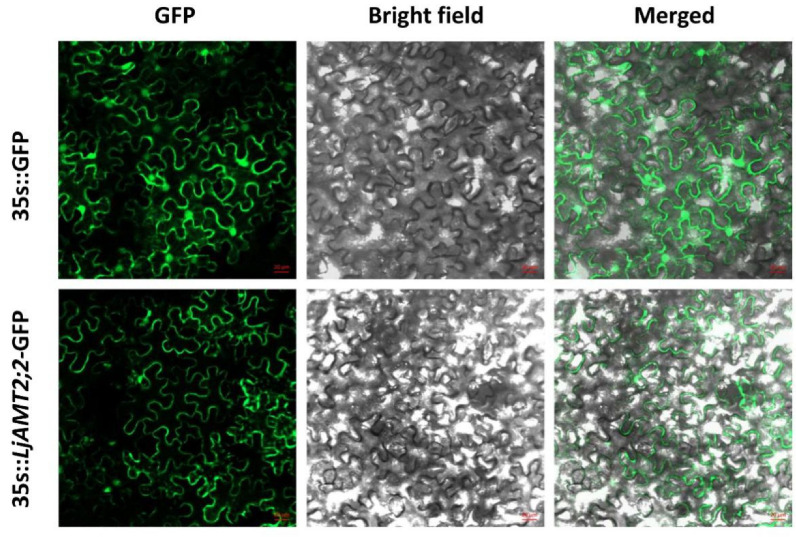
Subcellular localisation of *LjAMT2;2* expression. 35s::GFP is an empty carrier, 35s::*LjAMT2;2*-GFP is the target gene. Three visual fields were observed: Green channel (GFP), bright channel (bright-field), and merged. Scale bars, 20 μM.

**Figure 3 ijms-23-09522-f003:**
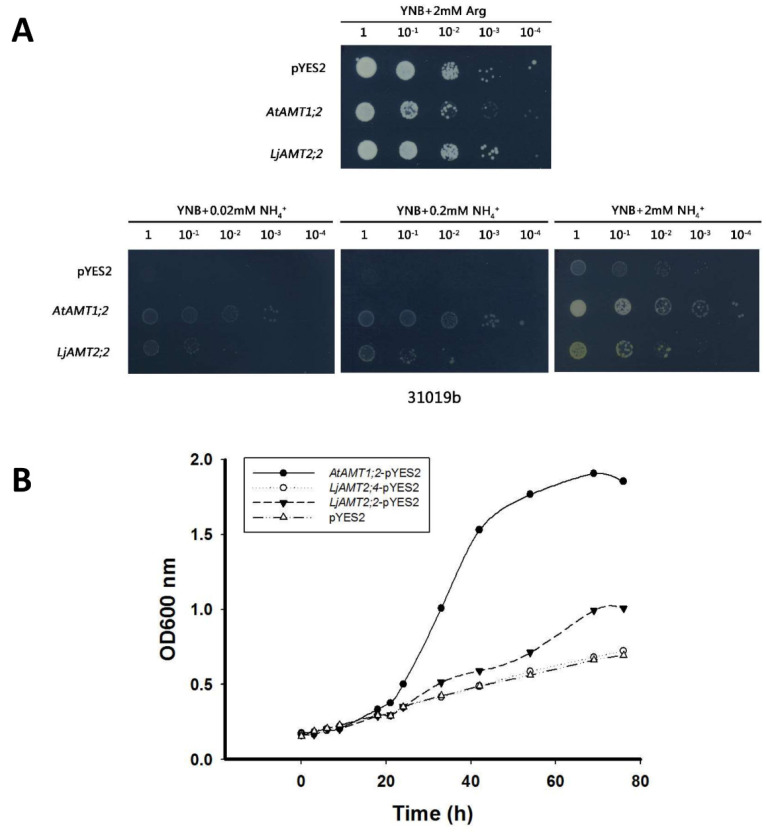
Functional analysis of *LjAMT2;2* in ammonium-deficient yeast. (**A**) Ammonium-deficient yeast supplementation experiment. (**B**) Yeast growth curves.

**Figure 4 ijms-23-09522-f004:**
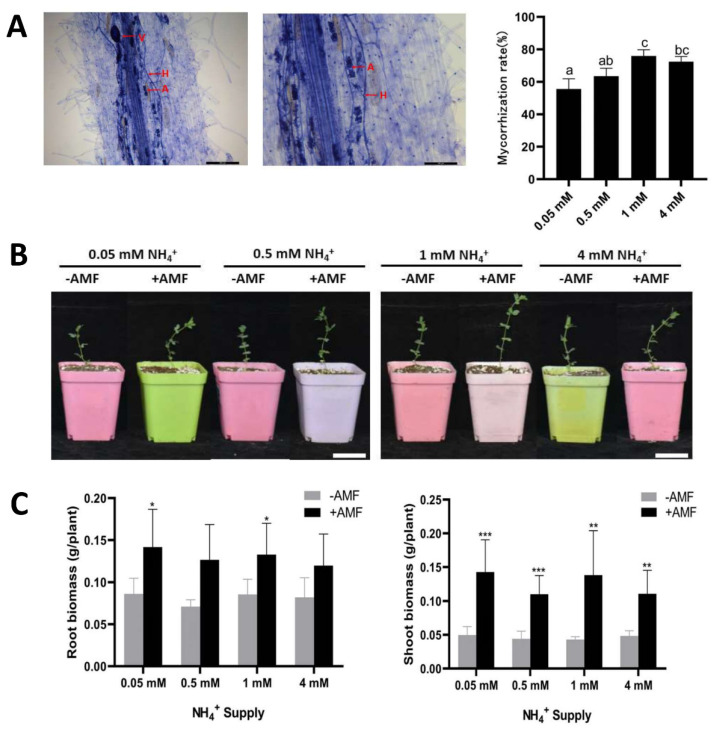
Effects of AMF colonisation on the growth of *Lotus japonicus* under different ammonium concentrations. (**A**) Trypan blue staining and colonisation rate statistics. H, Hyphae; V, vesicle; A, arbuscule (**B**) Phenotypes of wild-type plants under different ammonium ion concentrations. (**C**) Biomasses of aboveground and underground parts of plants under symbiotic and non-symbiotic conditions. Data are means ± standard error (SE; n ≥ 3 biological replicates; * *p* ≤ 0.05, ** *p* ≤ 0.01, *** *p* ≤ 0.001).

**Figure 5 ijms-23-09522-f005:**
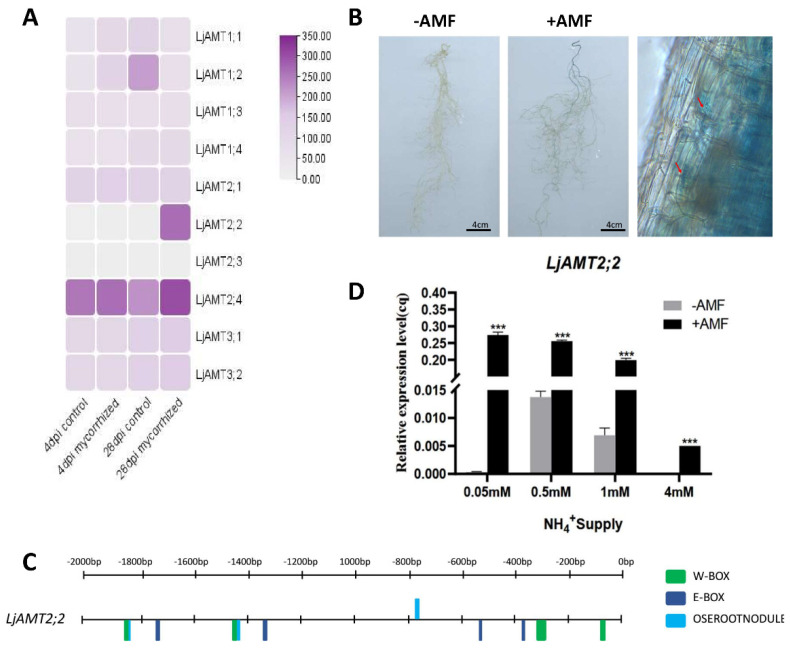
Mycorrhizal-inducible expression of the *LjAMT2;2* gene. (**A**) Heatmap of the LjAMT family gene expression induced by AMF. (**B**) GUS expression induced by AMF in p*LjAMT2;2* plants. Arrows indicate Gus expression positions (**C**) Analysis of cis-acting elements in the *LjAMT2;2* gene promoter. (**D**) Expression of the *LjAMT2;2* gene under symbiotic and non-symbiotic conditions. Data are means ± standard error (SE; n ≥ 3 biological replicates; *** *p* ≤ 0.001).

**Figure 6 ijms-23-09522-f006:**
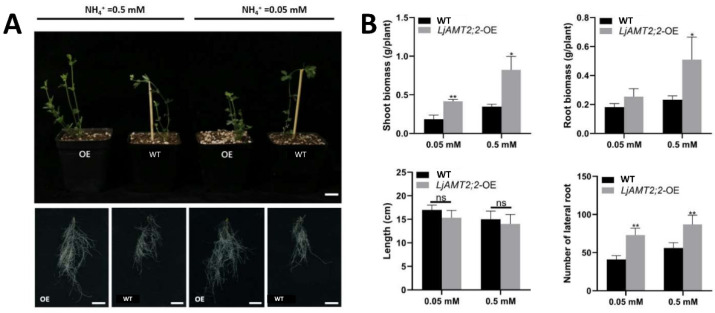
Comparison of phenotypes between LjAMT2;2-OE and wild-type plants. (**A**) Growth status of aboveground and underground parts of plants. (**B**) Biomass statistics and LjAMT2;2 gene expression levels. Data are means ± standard error (SE; n ≥ 3 biological replicates; * *p* ≤ 0.05, ** *p* ≤ 0.01, ns: no significant difference).

**Figure 7 ijms-23-09522-f007:**
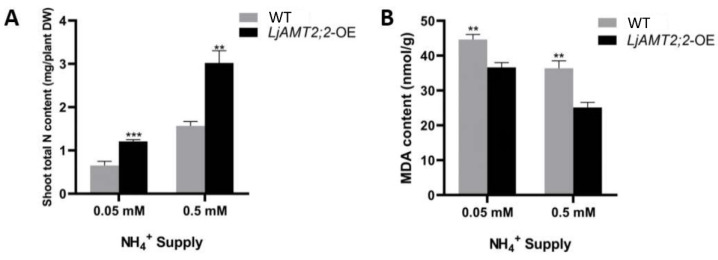
Comparison of physiological indices between *LjAMT2;2*-OE and wild-type plants. (**A**) Total nitrogen content. (**B**) MDA content. Data are means ± standard error (SE; n ≥ 3 biological replicates; ** *p* ≤ 0.01, *** *p* ≤ 0.001).

## Data Availability

The data presented in this study are available in article or Appendix A.

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
