# Peer review of "LjAMT2;2 Promotes Ammonium Nitrogen Transport during Arbuscular Mycorrhizal Fungi Symbiosis in Lotus japonicus"

_ijms, 2022, doi:10.3390/ijms23179522_

Round 1
Reviewer 1 Report
This investigation is about functional and structural characteristics of ammonium transporter LjAMT2;2 under symbiosis with arbuscular mycorrhizal fungi. Authors took interesting results about involvement of LjAMT2;2 in ammonium nitrogen transport and as a consequence its influence on plant growth.
However, some parts should be improved. The study is about Lotus japonicas transporter, that is why it is good to add “Lotus japonicas” in keywords.
Introduction should be rewritten. The study is about functioning of ammonium transporter, not about AMF. LjAMT2;2 has been studying since 2009 year. There are a lot of information about it. And according to your introduction one might suppose it has not been investigated at all, because there is no any word about this transporter in introduction. There is a part about transporters in other plants, but then suddenly the speech turns to what has been done in the present work. And it stays unclear what was the reason to study functioning LjAMT2;2. Moreover there are LjAMT2;1, LjAMT2;3 and LjAMT2;4. Why did you decided to investigate LjAMT2;2? There are some words in discussion but it should be in introduction and it should give more information. Also on Figure 3 B there are results for LjAMT2;4, but it has not been mentioned in the text of results or in discussion at all.
Line 170 – Lotus japonicus needs Italic
figure S1 A – I suppose 35S is enhanced
figure S1 B – why do you compare expression of LjAMT2;2-OE with Gifu, if you used MG20 as wild-type?
In my opinion, there is no necessity to mention the name of wild-type line everywhere in captions for figures (figure 4). Only one wild-type line was used, the name is said in materials and methods and this is enough. In pictures, it is possible to write – WT (wild-type) (figure 6).
Line 301 – something is wrong with the reference Yoshihiro Kobae
Line 363 – what was the line or cultivar of Nicotiana benthamiana?
Line 368 – brackets for MG20 are unnecessary
Line 369 – check the correct name of fungus, also it has to be italic
Part 4.5 is unclear. Lines 377-378 are about bacteria, and then 2 days of germination. Germination of what? It seems something is missed.
Line 407- write the country for Transegene.
Line 409 – write the country for Vazyme and check the name of SYBRGREEN kit
Author Response
Dear Reviewer #1,
Thank you very much for your constructive comments and we have revised the manuscript accordingly. Below please find our point-to-point response.
Point 1: This investigation is about functional and structural characteristics of ammonium transporter LjAMT2;2 under symbiosis with arbuscular mycorrhizal fungi. Authors took interesting results about involvement of LjAMT2;2 in ammonium nitrogen transport and as a consequence its influence on plant growth. However, some parts should be improved. The study is about Lotus japonicas transporter, that is why it is good to add “Lotus japonicas” in keywords.
Response 1: We thank the reviewer for the positive comments. We have added “Lotus japonicas” in keywords.
Point 2: Introduction should be rewritten. The study is about functioning of ammonium transporter, not about AMF. LjAMT2;2 has been studying since 2009 year. There are a lot of information about it. And according to your introduction one might suppose it has not been investigated at all, because there is no any word about this transporter in introduction. There is a part about transporters in other plants, but then suddenly the speech turns to what has been done in the present work. And it stays unclear what was the reason to study functioning LjAMT2;2. Moreover there are LjAMT2;1, LjAMT2;3 and LjAMT2;4. Why did you decided to investigate LjAMT2;2? There are some words in discussion but it should be in introduction and it should give more information. Also on Figure 3 B there are results for LjAMT2;4, but it has not been mentioned in the text of results or in discussion at all.
Response 2: We thanks the reviewer for these excellent comments. We have extensively revised the Introduction as you suggested.
- We have included the LjAMT2;2 references as follows: “Indeed, it has been reported that in the transcriptomic analysis of Lotus japonicus roots after AMF colonization, LjAMT2;2 was shown to be the most up-regulated gene, and the transcript is preferentially located in arbuscular cells and has the function of transporting NH3, which is the key to AMT in mycorrhiza.”
- The reason why we choose LjAMT2;2 has been added in the revised Introduction as follows: “Heatmap analysis of transcriptome data from japonicus under symbiosis formation for 4 and 28 days and non-symbiotic conditions showed that expression of AMT1 subfamily genes was hardly affected by AMF. By contrast, AMF had a significant effect on expression of AMT2 family members, consistent with previous reports on AMTs induced by AMF in other species, among which the LjAMT2;2 gene was most significantly affected by AMF.”
- For the LjAMT2;4 in the Figure 3B, we have added a sentence to describe its limited role in ammonium transportation as follows: “Meanwhile, our results showed another AMT2 gene in L. japonicus, LjAMT2;4, had limited function in transporting ammonium nitrogen”.
Point 3: Line 170 – Lotus japonicus needs Italic
Response 3: We have revised accordingly.
Point 4: figure S1 A – I suppose 35S is enhanced
Response 4: Yes, to ensure the most overexpression effect of LjAMT2;2, we used the enhanced 35S promoter. We have added the information in Materials and Methods.
Point 5: figure S1 B – why do you compare expression of LjAMT2;2-OE with Gifu, if you used MG20 as wild-type?
Response 5: We thank the reviewer for pointing out the confusing mistake. It was MG20 as wild-type control, and we have corrected it the revised supplement.
Point 6: In my opinion, there is no necessity to mention the name of wild-type line everywhere in captions for figures (figure 4). Only one wild-type line was used, the name is said in materials and methods and this is enough. In pictures, it is possible to write – WT (wild-type) (figure 6).
Response 6: Thanks for the good suggestion and we have revised accordingly. The Figure 6 is modified as follows:
Point 7: Line 301 – something is wrong with the reference Yoshihiro Kobae
Response 7: We thank the reviewer for pointing out the confusing mistake. The references have been revised.
Point 8: Line 363 – what was the line or cultivar of Nicotiana benthamiana?
Response 8: The cultivar of Nicotiana benthamiana was LAB.
Point 9: Line 368 – brackets for MG20 are unnecessary
Response 9: The brackets were removed.
Point 10: Line 369 – check the correct name of fungus, also it has to be italic
Response 10: We have revised the names accordingly.
Point 11: Part 4.5 is unclear. Lines 377-378 are about bacteria, and then 2 days of germination. Germination of what? It seems something is missed.
Response 11: Sorry for the confusing statements. We have revised this section by adding the following sentence: “The verified positive colonies were transferred to YMB solid medium and cultured at 28°C for 2 days. During the culture of Agrobacterium tumefaciens, the seeds of L. japonicus were sterilized and germinated. After 2 days of germination of L. japonicus, ……”.
Point 12: Line 407- write the country for Transgen.
Response 12: We have added the country name (China). TransGen, China
Point 13: Line 409 – write the country for Vazyme and check the name of SYBRGREEN kit
Response 13: We have added the country name (China) for ChamQ Universal SYBR qPCR Master Mix (Vazyme, China).
Again, we thank the reviewer for these excellent suggestions.
Best regards,
Xiaoyu Li

Reviewer 2 Report
Dear authors, the idea of your research is interesting and some suggestions will improve your work.
Make a more impersonal text - remove "we", "our" etc in the entire manuscript and rewrite the sentences
Introduction is clear. I have one suggestion here - try to connect better (a sentence or a paragraph) to clearly say why the LjAMT2;2 is important for agriculture. Add a paragraph where to discuss about other researches on LjAMT2;2 - if there are many or is a new research.
Figure 2 - write a caption of this figure - what means a), b), c) ... from the figure is clear, but you need to explain. Add in each figure from the manuscript what the codes in the images means. This will make each caption independent from the rest of the text.
Results and Discussion sections are clear, only the figure caption should be improved.
Materials and methods - pay attention to species name (line 369 - Glomuintraradices) and say the source of biological material used
Add
Author Response
Dear Reviewer #2,
Thank you very much for your constructive comments and we have revised the manuscript accordingly. Below please find our point-to-point response.
Point 1: Make a more impersonal text - remove "we", "our" etc in the entire manuscript and rewrite the sentences.
Response 1: Thanks for your suggestion. The corresponding sentences have been revised in the manuscript.
Point 2: Introduction is clear. I have one suggestion here - try to connect better (a sentence or a paragraph) to clearly say why the LjAMT2;2 is important for agriculture. Add a paragraph where to discuss about other researches on LjAMT2;2 - if there are many or is a new research.
Response 2: We thanks the reviewer for the excellent comments. We have included the LjAMT2;2 references as follows: “Indeed, it has been reported that in the transcriptomic analysis of Lotus japonicus roots after AMF colonization, LjAMT2;2 was shown to be the most up-regulated gene, and the transcript is preferentially located in arbuscular cells and has the function of transporting NH3, which is the key to AMT in mycorrhiza. Heatmap analysis of transcriptome data from L. japonicus under symbiosis formation for 4 and 28 days and non-symbiotic conditions showed that expression of AMT1 subfamily genes was hardly affected by AMF. By contrast, AMF had a significant effect on expression of AMT2 family members, consistent with previous reports on AMTs induced by AMF in other species, among which the LjAMT2;2 gene was most significantly affected by AMF.”
Point 3: Figure 2 - write a caption of this figure - what means a), b), c) ... from the figure is clear, but you need to explain. Add in each figure from the manuscript what the codes in the images means. This will make each caption independent from the rest of the text. Results and Discussion sections are clear, only the figure caption should be improved.
Response 3: We thank the reviewer for the comments. We have add “35s::GFP is an empty carrier, 35s::LjAMT2;2-GFP is the target gene. Three visual fields were observed (GFP, Bright field, Merged).” in caption.
Point 4: Materials and methods - pay attention to species name (line 369 - Glomuintraradices) and say the source of biological material used
Response 4: We thank the reviewer for pointing out the mistake. The AMF species was Glomus intraradices (Gi), which was obtained from Sun Yat-Sen University.
Thanks again to the reviewer for professional suggestions.
Best regards,
Xiaoyu Li

Round 2
Reviewer 1 Report
Introduction is good, but some parts are repeated. Lines 86-116 has to be deleted. They repeat lines 64-80. Where is the 25th reference?
Line 73 – reference 28 is in the middle of the sentence.
Line 72 – references 26, 22, 23, 27, 28 are at the beginning of the sentence.
Line 118 - references 31-34 are at the beginning of the sentence
Figure 2. in a caption, it is better to say green channel – GFP, grey channel – bright field. Howere, in this case it is clear what is green and what is grey.
Line 212 - Lotus japonicus needs Italic
Figure 6 A and figure 7 – in a caption “wild type” is used, so the same term should be used in a picture.
Line 449 – give the name of laser confocal microscope, with manufacture and the country
Linw 453 – “MG20and”
Line 454 – give the city and a country of the university. What is the name of the culture medium?
Line 466 – how seed were sterilized? At least give the chemical and time.
Line 513 – manufacturer is Solarbio, also add the country
Figure S1A - 35S is enhanced. Check the spelling
Figure S1B - MG20 should be repleced with WT. According to the picture it is not clear what is MG20
Author Response
Dear Reviewer #1,
Thank you very much for your constructive comments and we have revised the manuscript accordingly. Below please find our point-to-point response.
Point 1: Introduction is good, but some parts are repeated. Lines 86-116 has to be deleted. They repeat lines 64-80. Where is the 25th reference?
Point 2: Line 73 – reference 28 is in the middle of the sentence.
Point 3: Line 72 – references 26, 22, 23, 27, 28 are at the beginning of the sentence
Point 4: Line 118 - references 31-34 are at the beginning of the sentence
Response: Sorry for the confusing "track changes" manuscript. These points won’t be the problem once we remove “track changes”.
Point 5: Figure 2. In a caption, it is better to say green channel – GFP, grey channel – bright field. Howerer, in this case it is clear what is green and what is grey.
Response: We thank the reviewer for the comments. We have added “Three visual fields were observed (Green channel-GFP, Bright channel-Bright field, Merged) .” in caption.
Point 6: Line 212 - Lotus japonicus needs Italic
Response: We have modified it accordingly.
Point 7: Figure 6 A and figure 7 – in a caption “wild type” is used, so the same term should be used in a picture.
Response 7: Thanks for the good suggestion and we have revised accordingly. The Figures are modified as follows:
Point 8: Line 449 – give the name of laser confocal microscope, with manufacture and the country
Response: We have added the manufacture and the country (Leica LSM800,Germany).
Point 9: Line 453 – “MG20and”
Response: The two words have been separated.
Point 10: Line 454 – give the city and a country of the university. What is the name of the culture medium?
Response: We have added the country “Sun Yat-Sen University, Guangzhou, China.”. The culture medium has been changed to sand/soil mixed culture medium.
Point 11: Line 466 – how seed were sterilized? At least give the chemical and time.
Response: We have added the description in Materials and Methods: “Seeds were sterilized with 12% NaClO for 10min; washed with 75% ethanol for 3 times, 1-2 min each time; washed with sterile water for 3-5 times, 4-5 min each time.”
Point 12: Line 513 – manufacturer is Solarbio, also add the country
Response 12: We have added the country name (China). Solarbio, China
Point 13: Figure S1A - 35S is enhanced. Check the spelling
Response: Sorry for the mistake, we have revised Figure S1A.
Point 14: Figure S1B - MG20 should be repleced with WT. According to the picture it is not clear what is MG20
Response: We have revised it accordingly.
Again, we thank the reviewer for these excellent suggestions.
Best regards,
Xiaoyu Li
